# Treatment of Critical Size Femoral Bone Defects with Biomimetic Hybrid Scaffolds of 3D Plotted Calcium Phosphate Cement and Mineralized Collagen Matrix

**DOI:** 10.3390/ijms23063400

**Published:** 2022-03-21

**Authors:** Anna Carla Culla, Corina Vater, Xinggui Tian, Julia Bolte, Tilman Ahlfeld, Henriette Bretschneider, Alexander Pape, Stuart B. Goodman, Michael Gelinsky, Stefan Zwingenberger

**Affiliations:** 1University Center of Orthopedic, Trauma and Plastic Surgery, University Hospital Carl Gustav Carus at TU Dresden, 01307 Dresden, Germany; annacarla.culla@gmail.com (A.C.C.); corina.vater@uniklinikum-dresden.de (C.V.); julia.bolte@uniklinikum-dresden.de (J.B.); henriette.bretschneider@uniklinikum-dresden.de (H.B.); alexander.pape@uniklinikum-dresden.de (A.P.); stefan.zwingenberger@uniklinikum-dresden.de (S.Z.); 2Center for Translational Bone, Joint and Soft Tissue Research, University Hospital Carl Gustav Carus at TU Dresden, 01307 Dresden, Germany; tilman.ahlfeld@tu-dresden.de (T.A.); michael.gelinsky@tu-dresden.de (M.G.); 3University Center of Orthopaedics and Traumatology, University of Pisa, 56126 Pisa, Italy; 4Department of Orthopaedic Surgery, Stanford University, Stanford, CA 94305, USA; goodbone@stanford.edu

**Keywords:** 3D plotting, calcium phosphate cement, mineralized collagen, scaffold, osseointegration, bone tissue engineering

## Abstract

To treat critical-size bone defects, composite materials and tissue-engineered bone grafts play important roles in bone repair materials. The purpose of this study was to investigate the bone regenerative potential of hybrid scaffolds consisting of macroporous calcium phosphate cement (CPC) and microporous mineralized collagen matrix (MCM). Hybrid scaffolds were synthetized by 3D plotting CPC and then filling with MCM (MCM-CPC group) and implanted into a 5 mm critical size femoral defect in rats. Defects left empty (control group) as well as defects treated with scaffolds made of CPC only (CPC group) and MCM only (MCM group) served as controls. Eight weeks after surgery, micro-computed tomography scans and histological analysis were performed to analyze the newly formed bone, the degree of defect healing and the activity of osteoclasts. Mechanical stability was tested by 3-point-bending of the explanted femora. Compared with the other groups, more newly formed bone was found within MCM-CPC scaffolds. The new bone tissue had a clamp-like structure which was fully connected to the hybrid scaffolds and thereby enhanced the biomechanical strength. Together, the biomimetic hybrid MCM-CPC scaffolds enhanced bone defect healing by improved osseointegration and their differentiated degradation provides spatial effects in the process of critical-bone defect healing.

## 1. Introduction

The treatment of critical size bone defects is still a challenge for orthopedics [1,2]. Due to the limited amount and its donor site complications of autologous bone, bone substitutes have become an indispensable part of orthopedic surgery [3,4]. For repair of bone defects, composite biomaterials have comprehensive advantages in improving the biological characteristics due to the limitations of a single material in terms of biological, physical and chemical properties [4]. The combination of organic polymers and inorganic minerals is a promising approach to improve mechanical properties [4]. Natural bone tissue is mainly composed of organic collagen and inorganic hydroxyapatite (HA) [5,6]. From the biomimetic perspective, many organic/inorganic bone composites have been developed for bone tissue engineering, in which the organic part includes poly(citric acid), gelatin, collagen, chitosan, poly(glycolic acid) (PGA), poly(lactic acid) (PLA), poly(ε-caprolactone) (PCL) and their respective copolymers, and the inorganic part includes calcium phosphate, HA or bioactive glasses [4,6]. These composite materials combine the properties of both components, and show good biocompatibility and osteoconductivity [6,7]. Combining 3D images and CT data analysis, scaffolds with controllable structure, porosity and performance can be manufactured using 3D printing to match specific bone defects [4].

Calcium phosphate cement (CPC) is a synthetic bone substitute invented in 1986 [8]. CPC was approved by the Food and Drug Administration (FDA) for the treatment of non-weight-bearing bone defects in 1996 [3,8]. CPC is composed of calcium phosphate which can be converted into nanocrystalline HA when mixed with liquid to simulate natural bone mineralization structure and this process occurs under isothermal and physiological pH without surrounding tissue damage [3,9,10]. CPC has a promising future in clinical applications due to its superior properties including biocompatibility, osteoconductivity, injectability, moldability and biomechanical stability [3,11]. To make a personalized scaffold that meets the needs of the patient-specific bone defect, the CPC in this study consists of a calcium phosphate precursor (mainly α-tricalcium phosphate) and a biocompatible but hydrophobic (oil-based) carrier fluid that allows customized extrusion-based additive manufacturing (3D plotting) [9,10]. At present, CPC still has some shortcomings that limits its application, such as low degradation rate, slow osseointegration and weak biological response [11,12]. The current research is dedicated to improve osteogenesis and osseointegration to promote the application of CPC in bone tissue engineering [11]. Osseointegration is a process that involves implants and adjacent bone matrix, and includes osteogenesis and bone remodeling [13]. Incomplete osseointegration of bone material can be associated with serious postoperative complications including nonunion, implant failure and infection and subsequent pain and disability [13,14,15]. While seeding CPC scaffolds with undifferentiated rat mesenchymal stromal cells (MSC) did not increase bone formation in a cleft alveolar osteoplasty model in rats [9], immobilizing the cells within the CPC scaffolds using fibrin gel resulted in significant bone formation at the edge of the defect [10]. Therefore, the combination of CPC with a biomaterial that provides structural support for cells and enables them to colonize the scaffold might accelerate bone regeneration and expand its biological functions [3]. Biodegradable mineralized collagen matrix (MCM) can mimic the structure and composition of extracellular matrix of healthy bone tissue [16,17,18]. Scaffolds made of MCM by freeze-drying provide a spongy like structure with interconnective pores supporting deep and homogenous cell-seeding in vitro as well as quick tissue ingrowth and bone formation in vivo [16,17,18]. However, MCM is degraded quickly and might not provide effective biomechanical support and long-term biological effects [18]. Combining CPC and MCM can potentially overcome the deficiencies of each material separately, and create a composite material with distinct advantages biologically and mechanically.

In this study, hybrid scaffolds were made by 3D plotting a CPC scaffold which was filled afterwards with MCM. The hypothesis of this study is that the hybrid MCM-CPC scaffolds accelerate osseointegration and thereby promote the repair of a critical bone defect.

## 2. Results

Figure 1 provides an overview about the experimental work conducted in this study. MCM, CPC and MCM-CPC scaffolds were prepared and subsequently analyzed regarding their microstructure by a scanning electron microscopy (SEM). For the in vivo study, 48 male Wistar rats (age 12 weeks) were randomized into 4 groups: (1) control (CON, empty defect), (2) MCM (only mineralized collagen matrix), (3) CPC (only CPC) and (4) MCM-CPC (hybrid made of CPC filled with MCM). A 5 mm bone defect was created at the right femur of the rats, scaffolds were implanted or the defect was left empty (CON group). After an observation period of 8 weeks, animals were euthanized and femora were explanted to perform micro-computed tomography (µCT) scans, biomechanical testing and histological analysis.

### 2.1. Preparation of the Scaffolds

Macroporous CPC scaffolds were produced by 3D plotting with a cylindrical shape flattened at one side (Ø 4 mm, height = 4.5 mm, strand width = 800 µm, strand geometry = 60°) whereas microporous MCM scaffolds were prepared by freeze-drying (Ø 4 mm, height = 5 mm, mean pore size = 200 µm). To obtain the MCM-CPC hybrid scaffolds, plotted CPC scaffolds were filled with the MCM solution and then freeze-dried. From cross and sagittal sections of the MCM-CPC scaffolds it can be seen that the inner pores of the plotted CPC scaffold were evenly filled with MCM (Figure 2C–F). The sagittal view of SEM images showed that the CPC scaffold (Figure 3A–C) had a dense structure where the inner pores were interconnective. In line with the macroscopic images of the MCM-CPC, SEM images also show that the MCM is homogeneously distributed within the pores of the CPC scaffold and provide a three-dimensional structure with high porosity (Figure 3D–F). Since the MCM was just filled into the CPC scaffold and then freeze-dried, the boundary between MCM and CPC was clear with no strong connection between the two materials (Figure 3F).

### 2.2. In Vivo Study

#### 2.2.1. µCT Analysis 

In all groups the two-end regular shape of the bone defects once created by the osteotomy was lost and became conical or irregular shaped. In the CON and MCM group, defect width was decreased, indicating that there was new bone formation at the bony ends. Nevertheless, there was no complete bone bridge connecting the defect margins in both groups. In the CPC and MCM-CPC group, bone defects were filled with new bone tissue growing into the scaffolds (Figure 4). Analysis of the region of interest (ROI) of the original bone defect area showed that bone volumes (BV) in the CPC and MCM-CPC group were significantly higher than in the CON and MCM group (*p* < 0.05). Combination of the CPC scaffold with MCM (MCM-CPC) led to higher BV values compared to the CPC alone, but this difference was not statistically significant. The bone mineral density (BMD) results demonstrated that CPC and MCM-CPC groups had significantly higher values than CON and MCM groups (*p* < 0.05). The BMD value of the hybrid MCM-CPC group was also higher than that of the CPC group, but this difference was not statistically significant (Appendix A). 

Since the density of CPC present in the CPC and MCM-CPC scaffolds is in the range of the analyzed BMD, it is not possible to distinguish between CPC and new bone based on the µCT data only. Nevertheless, there is a small difference between MCM-CPC and CPC indicating that the MCM-CPC scaffolds increased new bone formation in the bone defect area. 

#### 2.2.2. Biomechanical Testing

Of the four groups, only CPC and MCM-CPC fit the inclusion criteria (adequate defect bridging) for biomechanical 3-point bending testing. The test results showed that the maximum load that could be applied to MCM-CPC scaffolds was significantly (*p* < 0.05) increased compared to CPC scaffolds (Figure 5). 

#### 2.2.3. Histological Analysis 

Hematoxylin and eosin (H&E) and Masson–Goldner trichrome staining were conducted to analyze bone regeneration. In the CON group, cone-shaped new bone tissue was formed at both ends of the bone defect and the bone defect site was filled with fibrous tissue. In the MCM group, cone-shaped new bone tissue was formed at both ends of the bone defect. The MCM scaffold itself was almost completely degraded and only a small amount of MCM was still visible in the middle of the bone defect which was wrapped by fibrous tissue. In the CPC and MCM-CPC groups, new bone grew along the periphery and the inner pores of the CPC scaffold, which formed a clamp-like bone structure to fix the CPC scaffold within the defect. The non-degraded CPC scaffold was wrapped by fibrous tissue. In the CPC group, the 3-dimensional pores inside the scaffold were filled with new bone and fibrous tissue. In the MCM-CPC group, MCM inside the hybrid scaffolds was degraded rapidly which was consistent with the MCM-alone group. Thus, degradation of MCM allowed ingrowth of new bone and fibrous tissue. Compared with CPC alone, there was more new bone tissue that had been growing deeper inside the hybrid scaffolds. In all groups, bone defects were not completely connected by newly formed bone tissue (Figure 6).

The grade of defect healing was obtained according to the classification system of Huo et al. [19] by three independent observers. Compared to the CON group the grade of defect healing was higher for the biomaterial groups with highest scores for MCM-CPC (*p* < 0.05). No difference regarding the grade of defect healing could be seen between the MCM and CPC groups (Figure 7A). 

Because the µCT results cannot show the actual new bone regeneration as a result of similar densities of the bone and the CPC structures, histological sections were also used to assess the bone formation quantitatively (Figure 7B–D). The percentage of new bone formation area to the initial defect size showed that the CON and CPC-MCM groups had comparable new bone formation area percentage, which was higher than the CPC and MCM groups, and the CPC group had the smallest new bone area percentage. The results of percentage of maximum distance of new bone to the initial defect length and percentage of total distance of new bone to the initial defect length were consistent. These results showed that the biomaterial groups were higher than the CON group in which the CPC and MCM-CPC groups were higher than MCM group. The MCM-CPC had the highest percentage of new bone tissue growth distance.

#### 2.2.4. Activity of Osteoclasts

Tartrate resistant acid phosphatase (TRAP) staining was conducted to evaluate the activity of osteoclasts (TRAP-positive osteoclasts appear purple-red). 

In all groups, osteoclasts were found in the newly formed bone tissue. Inside the bone defect, osteoclasts were occasionally seen in the CON group and few osteoclasts were seen in the fibrous tissue or around the nondegraded biomaterial in the MCM group. In the CPC group, there were a few osteoclasts in the interconnecting pores on the outer side and more osteoclasts can be seen in the deep connected pores around the nondegraded biomaterial or in the fibrous tissue in the MCM-CPC group (Figure 8).

## 3. Discussion

The focus of this study was to develop biomimetic hybrid scaffolds by combination of CPC with MCM to accelerate osseointegration and bone regeneration of critical size bone defects. Therefore, 3D plotted, porous CPC scaffolds were filled with MCM and implanted into 5 mm femoral defects in rats. 

During an 8-week observation period, the MCM inside the hybrid scaffolds was degraded rapidly and induced new bone ingrowth. Together with the remaining CPC the newly formed bone fully filled the initial defect area. Compared with the other groups, the MCM-CPC scaffolds increased new bone formation with bone growing deeper inside the scaffolds. Thus, the connection between bone and biomaterial was tighter and led to improved biomechanical properties. 

CPC is a biomaterial based on calcium phosphate which is transformed into nanocrystalline hydroxyapatite (HA) resembling the mineral phase of natural bone tissue with high biocompatibility [9,10,12]. MCM consists of mineralized collagen fibers, strongly mimicking the structure and composition of extracellular matrix of natural bone [16,17,18]. To obtain hybrid scaffolds, 3D plotted CPC scaffolds were simply filled with a MCM solution and freeze-dried which did not change the chemical properties of both materials. Since CPC and MCM alone show good biocompatibility [18,20], no adverse effects were expected for hybrid scaffolds.

Osseointegration in bone regeneration is essential for preventing implant failure and enhancing its function [21,22,23]. The osseointegration process involves multiple factors, including the host cells, the biomaterial type and physicochemical characteristics of implants and innate immune processes, which together participate in bone remodeling [13,21]. The regeneration of tissue defects is completed by cells and biomaterials filling the defect volume and providing a support structure for cell ingrowth [9]. Thus, bone cell ingrowth into the scaffold plays a crucial role in the process of bone regeneration and osseointegration. One of the cornerstones of the tissue engineering concept beside biomaterials and bioactive factors are cells. However, pre-seeding of implant materials is not always beneficial and also depends on either the type of cells, their differentiation status or both. As observed by Korn et al., in a rat cleft alveolar osteoplasty model, pre-seeding of 3D plotted CPC scaffolds with undifferentiated mesenchymal stromal cells (MSC) did not enhance defect healing [9]. Furthermore, in one of our previous studies, pre-seeding of MCM scaffolds with either undifferentiated dental pulp- or bone marrow-derived MSC did not enhance healing of critical bone defects in NMRI nude mice, although satisfactory results have been achieved in vitro [24]. 

The combination of CPC with MSC-laden fibrin increased bone formation [10], thus it seems more feasible to combine CPC with another material that can induce cells to grow into the scaffold to enhance osseointegration. Some researchers have proposed that the ideal bone grafts in the future may contain a combination of biomaterials with different characteristics, which can control mechanical properties, pore morphology, interconnectivity, surface structure and controlled biodegradation [25]. In the present study, the CPC scaffold was prepared by 3D printing which allows the production of customized and individualized implants. Thus, CPC strands with a strand-to-strand distance of 800 µm were rotated by 60° to create triangular-shaped pores. As seen in a previous study, 60° CPC strand rotation was favorable compared to a 30° regarding new bone formation [9]. To obtain hybrid scaffolds, printed CPC scaffolds were then filled with the MCM suspension. By freeze drying the constructs, the characteristic open porous MCM structure was formed right inside the plotted CPC showing no relevant physicochemical interaction between the materials. Since the hybrid MCM-CPC contains no cells, it can be sterilized and stored before implantation that eases a potential clinical application for non-weight bearing bone defects.

The bone defect in the CON group did not show signs of bridging according to µCT and histological analysis, confirming that the chosen femoral model was a critical size bone defect [2]. The newly formed bone had a cone-like shape in the CON and MCM groups, and a clamp-like structure in the CPC and MCM-CPC groups. The percentage of new bone formation area related to the initial defect size in the CON and MCM groups was comparable, but higher than in the CPC group. Similar results between CON and CPC groups were also observed in another artificial alveolar cleft study [9]. This can be explained by the enhanced bone formation at the defect ends. In the MCM group, rapid degradation of the MCM provided enough space for the new bone to grow, especially at the defect ends. In contrast, in the CPC containing groups the slow-degrading CPC hindered bone ingrowth and new bone was only able to grow along the periphery and the inner pores of the scaffold. Although the CON group showed increased area of the new bone, it could not completely repair the bone defect autonomously [1,2], and eventually the ends of the defect form a pseudarthrosis. The scaffold acts as a guide for the new bone so that the new bone quickly forms a bone-bridge to repair bone defects. Since there was no scaffold to guide bone cell growth in the CON group without biomaterial and in the MCM group with the rapid degradation of the biomaterial, the growth distance of new bone in the two groups was less than that of the CPC and MCM-CPC groups regardless of the percentage of the maximum distance or total distance. This suggests that the scaffold materials are actively involved in the bone regeneration process as cell and molecular carriers [4]. 

The MCM can mimic the extracellular matrix in bone which is mainly composed of HA, collagen I and other proteins. The interconnected pore structure and elasticity of MCM is thought to be advantageous concerning cell ingrowth and to ensure adequate contact between material and tissue [18]. The suitable pore size and interconnecting porosity of MCM can foster infiltration of cells [17] and increases the surface area when used in combination with the CPC which is conductive to cell adhesion. Both, collagen I and HA, can promote osteoblast differentiation and together they improve bone formation [3,26]. Although, there is only a small difference between the CPC and MCM-CPC groups regarding the volume of newly formed bone (BV) and bone mineral density (BMD), it seems as if the addition of MCM to the CPC scaffold further supports new bone formation. This can be verified by the results of the histomorphometric analysis where significantly higher values for maximum and total new bone tissue distance were observed for the MCM-CPC group. MCM promoted the repair of bone defects by accelerating the growth of new bone in the hybrid scaffolds. In line with the µCT and histomorphometric data, the biomechanical data also showed a higher mechanical strength for hybrid compared to pure CPC scaffolds indicating that the addition of MCM accelerated the osseointegration of the hybrid scaffolds. Compared to the CPC, the MCM does not add any significant mechanical stiffness and thus the higher mechanical strength must be attributed to better osseointegration of the MCM-CPC scaffold. This might be caused by an increased number of bone-to-scaffold contacts leading to more connections between bone and scaffold and thus a higher mechanical strength. 

For clinical applications implant materials usually need to be larger than for animal experiments. While small scaffolds may be fully colonized with cells, in large scaffolds there is the problem of insufficient nutrient and oxygen supply in the inner parts, especially when the implant materials are compact and dense. Thus, cells may grow only on the outer surface [10]. Creative 3D bioprinting can solve these problems of oxygen and nutrient supply and also allows functionalization of the scaffolds [27,28]. The ideal scaffold can temporarily replace natural tissues, interact with the surrounding microenvironment, and actively guide cellular events, ultimately leading to faster bone formation [4]. One possibility to further enhance the performance of 3D printed large scaffolds is the combination with additional cell-supporting materials, such as MCM in this study. 

Osteoclasts play an important role in the process of cell-mediated degradation of CPC by resorption and phagocytosis into intracellular vesicles [29,30]. Additionally, osteoclast-mediated osseointegration can be regulated by the calcium/phosphate ratio [13]. Compared to CPC, in the MCM-CPC group more TRAP-positive osteoclasts could be found in the pores of the scaffold and can accelerate the degradation of the CPC. In the present study no substantial degradation of the CPC could be observed within an observation period of 8 weeks. As shown by a previous study, even after 12 weeks printed CPC scaffolds were intact without distinct marks of degradation [31]. In the hybrid scaffolds, CPC and MCM both maintained their own degradation performance. This differentiated degradation characteristic between CPC and MCM provided a spatial effect for bone regeneration, which improved the drawbacks that the slow degradation of CPC cannot provide enough space and the fast degradation of MCM cannot provide a bridge in the process of bone regeneration. 

Considering the inhibitory effect of slow degradation of CPC in the initial stage of bone regeneration, the hard scaffold is designed to be concave at both ends and filled with soft scaffold, which may enhance the repair of the bone defect. Moreover, the good biocompatibility of MCM enables cell seeding with, e.g., human bone MSCs, human trabecular bone cells and other cells producing growth factors [32,33] that offers the opportunity of further evolution of these scaffolds, paving the way for a new evolution of critical size bone defect treatment. 

As with the majority of studies, the design of the current study is subject to limitations. Bone regeneration was observed only 8 weeks after surgery. In another study where the CPC was loaded with MSC-laden fibrin we found that even though the fibrin had been completely degraded at 6 weeks, a significant enhancement of bone regeneration was observed at 12 weeks post operation accompanied by significant degradation of the CPC [10]. This makes it possible that MCM-CPC hybrid scaffolds can further enhance bone repair in the following time accompanied by degradation of CPC under MCM-mediated cellular events. Then eventually, a bone bridge can be formed from both ends of the defect inside or along the scaffold to repair the defect. 

## 4. Materials and Methods

### 4.1. Preparation of the Scaffolds 

The CPC paste (INNOTERE, Radebeul, Germany) was sterilized by γ-irradiation (25 kGy) and then used for scaffold plotting. The CPC scaffold was fabricated according to previous work in our team [9,10]. Briefly, the CPC paste was plotted with a multichannel plotting system based on the BioScaffolder device from GeSiM (Großerkmannsdorf, Germany) at room temperature and with a cylindrical shape flattened at one side (Ø 4 mm, height = 4.5 mm, strand width = 800 µm, strand geometry = 60°). MCM scaffolds were produced according to previous work in our lab [17,18]. Briefly, collagen I (Syntacoll, Saal/Donau, Germany) was dissolved in 10 mM hydrochloric acid and mixed with a calcium chloride solution to adjust the pH to 7. For mineralization of the collagen, the solution was heated to 37 °C in the buffer for 12 h to allow the formation of collagen fibrils and the formation of nanocrystalline HA. The mineralized collagen was then mixed with the mother liquor and stirred until a newly cast suspension was formed, which was used for filling and placing in the cavity of a 48-well plate and frozen at −20 °C. The sample was freeze-dried and then cross-linked with 1 weight% N-(3-dimethylaminopropyl)-N-ethylcarbodiimide (EDC) hydrochloride in 80 vol.% ethanol for 1 h. The scaffolds were thoroughly rinsed in distilled water followed by a 1% glycine solution, and finally freeze-dried to obtain the MCM scaffold (Ø 4 mm, height = 5 mm, mean pore size = 200 µm). For MCM-CPC hybrid scaffolds, plotted CPC scaffolds were filled with the MCM solution and then freeze-dried. All scaffolds used for the in vivo study were sterilized by γ-irradiation (≥25 kGy) before use.

### 4.2. Animals

Forty-eight Wistar rats (male, 12-week-old, weight: 447.5 ± 80.5 g) were purchased from Janvier Labs (Le Genest-Saint-Isle, Mayenne department, France). The rats were randomized into 4 groups (Table 1) and fed a standard diet with food and water ad libitum with a 12-h light-and-dark cycle. All experiments were performed in adherence to the National Institutes of Health Guidelines for the Use of Experimental Animals and were approved by the Local Animal Care and Ethics Committee of Dresden University Hospital (protocol no. DD24-5131/354/10, 29 April 2016).

### 4.3. Surgical Procedure

Anesthesia was induced by inhalation of the mixture of isoflurane and O_2_ and it was maintained at a flow rate of 1.5–2 L/min (isoflurane vaporizer: Ohmeda-Isotec 3, BOC Health Care, Great Britain, UK). Analgesia was administered subcutaneously with buprenorphine (30 µg/kg) before starting the surgery.

Surgeries were performed as described previously [34]. Briefly, the rat was positioned in the prone position, and the right femur exposition was obtained by a 3 cm skin and fascia incision along the upper lateral thigh. A customized 5-hole internal fixation plate (locking plate, stainless steel; LCP Compact Hand System, Synthes GmbH, Oberdorf, Switzerland) was placed anterolaterally and fixed to the femur using 2 screws at the proximal and distal ends respectively (Ø 1.5 mm locking screws, stainless steel; length outer screws: 7 mm, length inner screws: 6 mm; Synthes GmbH). Whereupon a 5 mm defect was created by 2 Gigli wires (0.44 mm; RISystem AG, Landquart, Switzerland) using a 5 mm custom-made 3-dimensional printed saw guide. The bone tissue in the middle of the bone defect was carefully removed, and the wound was rinsed thoroughly with saline to remove bone chips and metal residues of the Gigli wires. The bone defect was filled with the group depending on scaffold (MCM, CPC, MCM-CPC) or left empty (control group). The muscular suture was executed with Vicryl 4-0 (muscles/fascia; Ethicon, Johnson & Johnson, New Brunswick, NJ, USA), and the skin was sutured with Ethilon 4-0 (skin; Ethicon). The animals were transferred back into their cage and regularly monitored until they were fully recovered (Appendix A).

### 4.4. µCT

After an observation period of 8 weeks, the animals were euthanized after inhalation of isoflurane followed by 100% CO_2_ for 20 min. Right femurs of each animal were explanted, detached from soft tissue and placed in 4% buffered formaldehyde for 24 h during which µCT scanning and measurements were performed. The right femur of each rat was scanned using a SCANCO vivaCT 75 (SCANCO Medical AG, Brüttisellen, Switzerland). The following settings were used: X-ray energy 55 kVp, projections 525 and pixel resolution 20 µm. A standardized 3-dimensional region of interest (ROI) was defined with a cylindrical shape of 6 mm length and 5 mm diameter and was aligned centrally between the inner 2 screws of the locked plate. Bone was defined to have a tissue density of more than 200 mg hydroxyapatite per cm^3^. The bone volume (BV in mm^3^) and bone mineral density (BMD in mg HA/cm^3^) parameters of the ROI were obtained by the software of the SCANCO vivaCT 75.

### 4.5. Biomechanical Testing

The samples that were used for biomechanical testing were thawed overnight at 4 °C and then placed at room temperature for 1 h before testing. The sample was placed on a custom-made 3-point bending fixture for biomechanical testing. The sample was loaded at a rate of 1 mm/s until it was broken, and the maximum force of bone defect breaking was used to measure the mechanical strength. The test method was consistent with our previous study [34].

### 4.6. Histological Analysis 

For histological analyses, specimens were fixed in 4% neutral buffered formaldehyde right after explantation. Dehydration was performed for 12 h using a Thermo Scientific Tissue Processor STP420 (Thermo Fisher Scientific, Waltham, MA, USA) following embedding in Technovit 9100n methyl methacrylate. Five sections per femur of about 30 μm thickness were cut in the sagittal plane and mounted on silane-coated slices. To evaluate the histomorphology of bone healing, slides were stained with hematoxylin and eosin (H&E; Merck, Darmstadt, Germany) and Masson–Goldner staining kit (Fluka, Sigma-Aldrich). The evaluation of all histological sections was performed using a Keyence BIOREVO BZ-700 microscope (Keyence, Neu-Isenburg, Germany).

Histological grading of fracture healing was scored based on H&E- and Masson–Goldner-stained sections according to the numerical scoring scheme established by Huo et al. [19]. The results were analyzed by 3 independent, blinded observers. In order to evaluate the influence of the scaffold on bone regeneration, the growth of new bone tissue was quantified based on H&E and Masson–Goldner stainings. Therefore, the percentage of the new bone formation area related to the initial defect size (Equation (1)), the percentage of the maximum distance of the new bone to the initial defect length (Equation (2)) and the percentage of the total distance of the new bone to the initial defect length (Equation (3)) were measured using Image J Version 1.53e software (National Institutes of Health, Bethesda, MD, USA) [9,10]. In addition, tartrate-resistant acid phosphatase (TRAP) staining (Sigma–Aldrich, St. Louis, MO, USA) was used to analyze the activity of osteoclasts.
percentage of new bone formation area to the initial defect size = (sum of the new bone formation area at both ends / initial bone defect size) × 100%(1)
percentage of maximum distance of new bone to the initial defect length = (maximum distance of new bone formation from one end / initial bone defect length) × 100%(2)
percentage of total distance of new bone to the initial defect length = (sum of the farthest distances of new bone tissue at both ends / initial bone defect length) × 100%(3)

### 4.7. Statistics

Data were analyzed using SPSS Statistics 20 statistical software (SPSS, Inc., Chicago, IL, USA). The normality of distribution of continuous variables was tested by one-sample Kolmogorov–Smirnov test. Continuous variables with normal distribution were presented as mean ± standard deviation (SD). Statistical significance was tested by one-way ANOVA, followed by LSD’s post hoc test for multiple comparisons. A value of *p* < 0.05 was considered significant. Non-continuous data not following the Gaussian distribution were presented as boxplots with median, and statistical significance was tested by the non-parametric Kruskal–Wallis test. For post hoc testing the individual treatment groups were tested applying Mann–Whitney test. The independent-samples *t*-test was used to compare the biomechanical test results. The level of significance was set at *p* < 0.05 (*).

## 5. Conclusions

A novel type of biomimetic hybrid MCM-CPC scaffold was fabricated by combining macroporous, 3D plotted CPC scaffolds with microporous MCM allowing the production of customized and storable implants. Combination of CPC with MCM enhanced the osseointegration of the hybrid scaffolds by accelerating the growth of new bone into the scaffolds also leading to an enhanced biomechanical stability. The differential degradation of MCM and CPC can provide a spatial effect for the MCM-CPC scaffolds in the process of bone defect repairing.

## Figures and Tables

**Figure 1 ijms-23-03400-f001:**
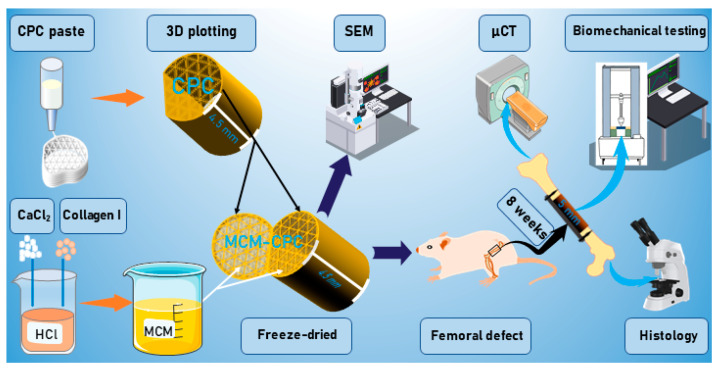
Scheme of the research and evaluation methods of this study (MCM: mineralized collagen matrix scaffold; CPC: calcium phosphate cement scaffold; MCM-CPC: hybrid scaffold consisting of 3D plotted calcium phosphate cement scaffold filled with mineralized collagen matrix; SEM: scanning electron microscopy; µCT: micro-computed tomography).

**Figure 2 ijms-23-03400-f002:**
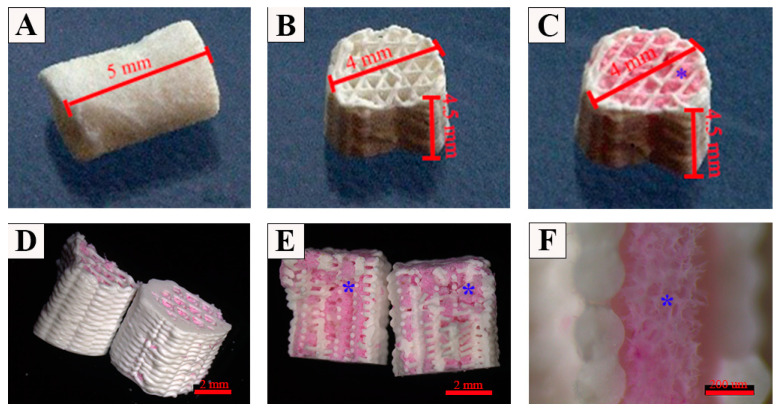
Appearance of scaffolds used for the treatment of the critical size femoral defects. (**A**) MCM scaffold; (**B**) CPC scaffold; (**C**–**F**) hybrid MCM-CPC scaffold (blue *: mineralized collagen matrix within 3D plotted CPC scaffolds stained with Sirius-red for better visualization).

**Figure 3 ijms-23-03400-f003:**
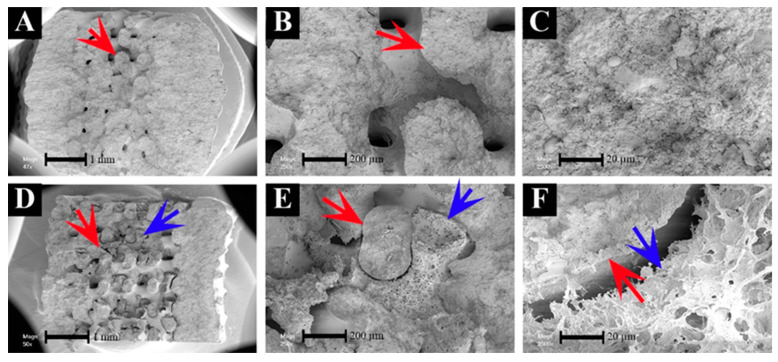
Scanning electron microscopy (SEM) images of CPC (**A**–**C**) and hybrid MCM-CPC scaffolds(**D**–**F**) at different magnifications (red arrows: CPC, blue arrows: MCM).

**Figure 4 ijms-23-03400-f004:**
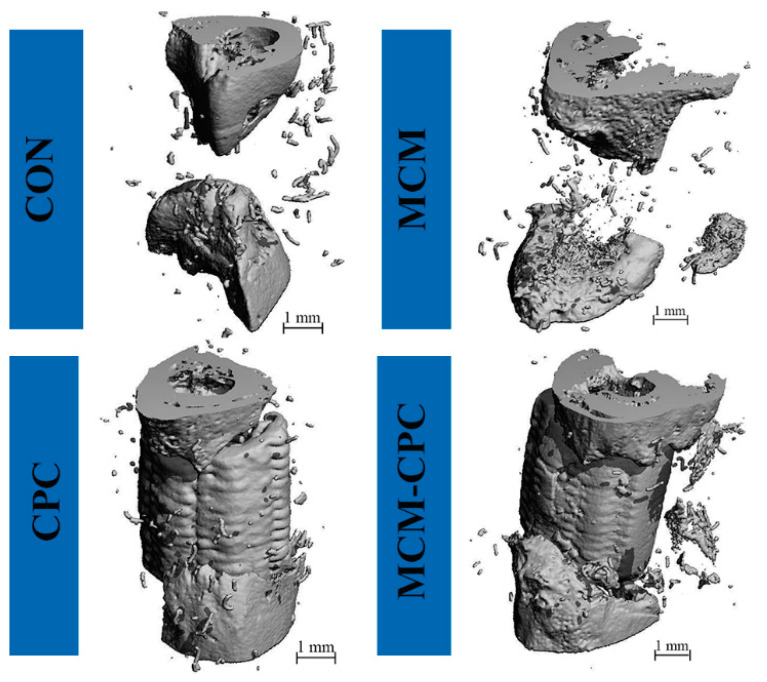
Representative examples of 3D reconstructions of the defect area (ROI) 8 weeks post-surgery as determined by µCT.

**Figure 5 ijms-23-03400-f005:**
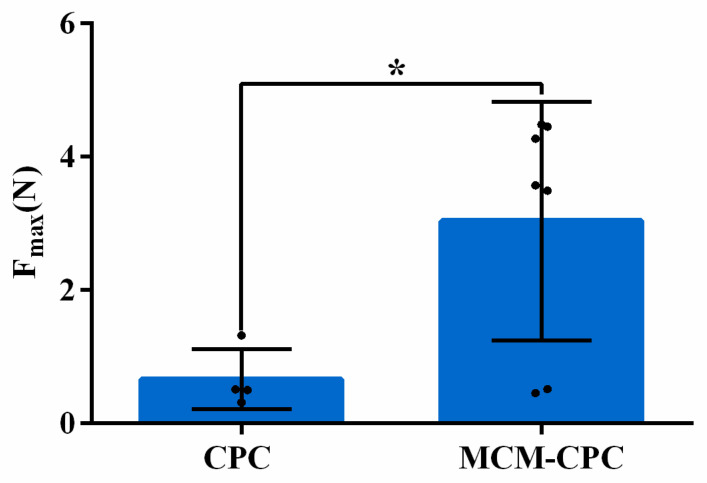
Maximum load applied to explants of the calcium phosphate cement (CPC) group and mineralized collagen matrix-calcium phosphate cement (MCM-CPC) group (F_max_: maximum force of bone defect breaking; mean ± SD, * *p* < 0.05).

**Figure 6 ijms-23-03400-f006:**
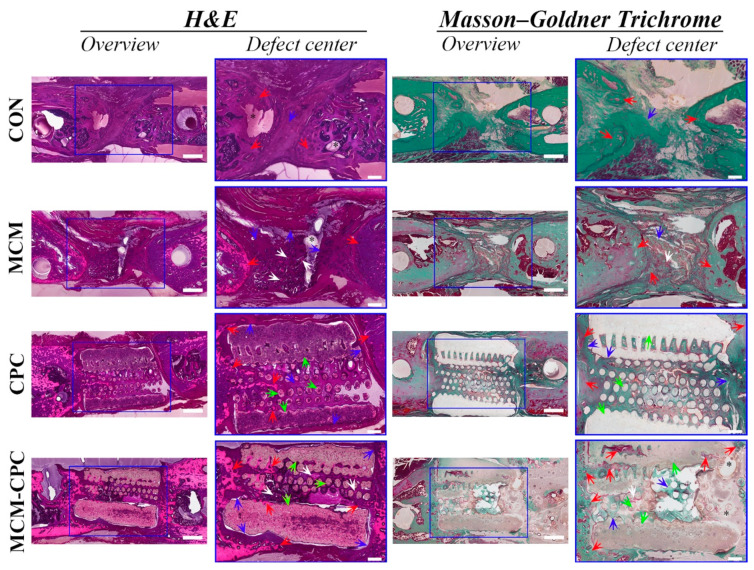
Representative H&E and Masson–Goldner stained sections of the defect area at week 8 weeks post-surgery (red arrows: bone, blue arrows: fibrous tissue, green arrows: remaining CPC, white arrows: remaining MCM; overview scale bars = 1 mm, defect center scale bars = 400 µm).

**Figure 7 ijms-23-03400-f007:**
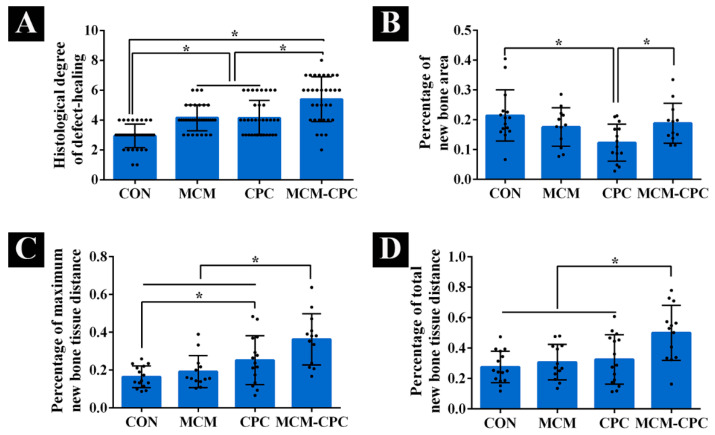
Histomorphometric analysis of new bone formation. (**A**) histological evaluation of the degree of defect healing according to Huo et al. [19] based on hematoxylin and eosin and Masson–Goldner-trichome stained sections, (**B**) percentage of new bone formation area related to the initial defect area, (**C**) percentage of maximum distance of new bone to the initial defect length, (**D**) percentage of total distance of new bone to the initial defect length (mean ± SD, * *p* < 0.05).

**Figure 8 ijms-23-03400-f008:**
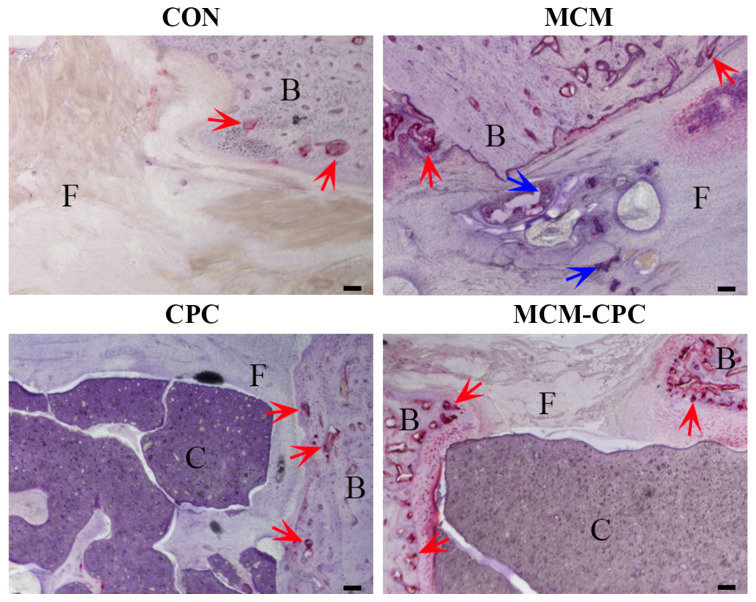
Representative tartrate resistant acid phosphatase (TRAP) stained sections of the defect area 8 weeks post-surgery (B: bone, F: fibrous tissue, C: remaining CPC; blue arrows: remaining MCM, red arrows: TRAP positive cells; scale bars = 100 µm).

**Table 1 ijms-23-03400-t001:** Study overview including treatment groups and sample size.

Group	Treatment	Animals Operated (*n*)	Animals for Analysis (*n*)
CON	empty defect without scaffold	12	12
MCM	mineralized collagen	12	11
CPC	3D printed calcium phosphate cement	12	12
MCM-CPC	3D printed calcium phosphate cement filled with mineralized collagen	12	12

## Data Availability

All data generated or analyzed during this study are included in this article.

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
