# Peer review of "Treatment of Critical Size Femoral Bone Defects with Biomimetic Hybrid Scaffolds of 3D Plotted Calcium Phosphate Cement and Mineralized Collagen Matrix"

_ijms, 2022, doi:10.3390/ijms23063400_

Round 1
Reviewer 1 Report
Summary
The study investigates an interesting approach to improve the treatment of critical size bone defects. A bone substitutes with comparable characteristics to autologous or allogenic bone is still missing and such research could prepare a way for the clinical application of an improved bone substitute.
The study applies sophisticated fabrication techniques to produce the samples and also sophisticated testing methods to investigate them. The methods and the results are well written and contain descriptive figures.
Major issues
The major issue relates to the description of a potential clinical application, which was somehow speculated in the discussion section. It should be pointed out that the application in rats is very different to the application in humans. The limitation concerns first the size and the related very slow degradation of the CPC scaffold, which negatively influence the complete “bridging” of the bone defect. Second, the brittleness of the CPC scaffold may prevent a clinical application in humans, since a crack in the scaffold will lead to non-healing of the bone. And third, also the metabolism is only conditionally comparable. Please mention such considerations in order to put a potential clinical application in the correct perspective.
Minor issues
The first minor issue relates to the definition of the four groups in the abstract. The definition of the groups is not obvious and should be rewritten.
Second, the description of the µCT method is not obvious. For me it is not clear how the bone volume was distinguished from scaffold volume only by a lower threshold. Furthermore, µCT results were presented and then it is mentioned that it is not possible to distinguish between CPC and new bone. Please clarify that.
Author Response
Response to Reviewer 1 Comments
Major issues
The major issue relates to the description of a potential clinical application, which was somehow speculated in the discussion section. It should be pointed out that the application in rats is very different to the application in humans. The limitation concerns first the size and the related very slow degradation of the CPC scaffold, which negatively influence the complete “bridging” of the bone defect. Second, the brittleness of the CPC scaffold may prevent a clinical application in humans, since a crack in the scaffold will lead to non-healing of the bone. And third, also the metabolism is only conditionally comparable. Please mention such considerations in order to put a potential clinical application in the correct perspective.
Special thanks to you for your valuable and constructive comments.
First, we totally agree with you that the application in rats is very different from the application in humans which was also mentioned in the discussion section. Our team expects and strives to improve the adverse effects of the CPC scaffold´s weak integration and low degradation rate regarding bone regeneration through process improvements or combining the CPC with other materials to generate hybrid scaffolds. The fragility of CPC scaffolds limits their clinical use and they are often used for non-weight-bearing bone defects. Considering that CPC has been approved by the FDA for clinical application, we just speculate the hybrid scaffolds in this study to be a potential clinical application. We know this requires more work to do and that is also what we are working towards in the future.
Minor issues
(1) The first minor issue relates to the definition of the four groups in the abstract. The definition of the groups is not obvious and should be rewritten.
Thank you for the suggestion. We have re-written the definition of the four groups in the abstract according to the reviewer’s suggestion.
(2) Second, the description of the µCT method is not obvious. For me it is not clear how the bone volume was distinguished from scaffold volume only by a lower threshold. Furthermore, µCT results were presented and then it is mentioned that it is not possible to distinguish between CPC and new bone. Please clarify that.
For the analysis of bone structure, the µCT images are binarized by setting a grey-level threshold. Above this threshold voxels are considered as bone and below this threshold voxels are considered as background1. Since the density of CPC present in the CPC and MCM-CPC scaffolds is similar to bone, it is not possible to distinguish between CPC and bone tissue by a lower threshold. Therefore, the values obtained by µCT analysis contain both – newly build bone and CPC. However, the CPC and MCM-CPC groups had the same CPC scaffold, so the difference could reflect the new bone tissue between the two groups. We cannot directly compare CPC containing groups (CPC and MCM-CPC group) with the no-CPC groups (CON group and MCM group). Therefore, we combined µCT analysis with histological analysis to observe the repair of bone defects.
Reviewer 2 Report
The present manuscript aims to develop biomimetic hybrid scaffolds by a combination of CPC with MCM to accelerate osseointegration and bone regeneration of critical size bone defects. The strengths of this study consist in the usage of a modern synthesis technique - 3D printing, and an extensive in-vivo evaluation of the bone defect repairing process. The authors have done a lot of experimental research, and the paper has a certain novelty and solid workload.
Author Response
The present manuscript aims to develop biomimetic hybrid scaffolds by a combination of CPC with MCM to accelerate osseointegration and bone regeneration of critical size bone defects. The strengths of this study consist in the usage of a modern synthesis technique - 3D printing, and an extensive in-vivo evaluation of the bone defect repairing process. The authors have done a lot of experimental research, and the paper has a certain novelty and solid workload.
Special thanks to you for reviewing our manuscript.